# A Review and Bibliometric Analysis on Applications of Microbial Degradation of Hydrocarbon Contaminants in Arctic Marine Environment at Metagenomic and Enzymatic Levels

**DOI:** 10.3390/ijerph18041671

**Published:** 2021-02-09

**Authors:** Gayathiri Verasoundarapandian, Chiew-Yen Wong, Noor Azmi Shaharuddin, Claudio Gomez-Fuentes, Azham Zulkharnain, Siti Aqlima Ahmad

**Affiliations:** 1Department of Biochemistry, Universiti Putra Malaysia, Serdang 43400, Selangor, Malaysia; gs58309@student.upm.edu.my (G.V.); noorazmi@upm.edu.my (N.A.S.); 2School of Health Sciences, International Medical University, Kuala Lumpur 57000, Malaysia; WongChiewYen@imu.edu.my; 3National Antarctic Research Center, Universiti Malaya, Kuala Lumpur 50603, Malaysia; 4Department of Chemical Engineering, Universidad de Magallanes, Avda. Bulnes 01855, Punta Arenas, Chile; claudio.gomez@umag.cl; 5Center for Research and Antarctic Environmental Monitoring (CIMAA), Universidad de Magallanes, Avda. Bulnes 01855, Punta Arenas, Chile; 6Department of Bioscience and Engineering, Shibaura Institute of Technology, Saitama-shi 337-8570, Saitama, Japan; azham@shibaura-it.ac.jp

**Keywords:** hydrocarbon degradation, microbial community, Arctic marine, review, bibliometric analysis

## Abstract

The globe is presently reliant on natural resources, fossil fuels, and crude oil to support the world’s energy requirements. Human exploration for oil resources is always associated with irreversible effects. Primary sources of hydrocarbon pollution are instigated through oil exploration, extraction, and transportation in the Arctic region. To address the state of pollution, it is necessary to understand the mechanisms and processes of the bioremediation of hydrocarbons. The application of various microbial communities originated from the Arctic can provide a better interpretation on the mechanisms of specific microbes in the biodegradation process. The composition of oil and consequences of hydrocarbon pollutants to the various marine environments are also discussed in this paper. An overview of emerging trends on literature or research publications published in the last decade was compiled via bibliometric analysis in relation to the topic of interest, which is the microbial community present in the Arctic and Antarctic marine environments. This review also presents the hydrocarbon-degrading microbial community present in the Arctic, biodegradation metabolic pathways (enzymatic level), and capacity of microbial degradation from the perspective of metagenomics. The limitations are stated and recommendations are proposed for future research prospects on biodegradation of oil contaminants by microbial community at the low temperature regions of the Arctic.

## 1. Introduction

Arctic crude oil resources have become the major interest for exploitation to power the petroleum and other related industries [1,2]. A growing number of areas are being exposed to anthropogenic activities, oil drilling and maritime traffic in the inclement Arctic climate, which pose a high risk for accidental oil spills. The Polar Code discourages the transport and use of heavy fuel oil (HFO) in the Arctic while being explicitly prohibited in the Antarctic under the International Convention for the Prevention of Pollution from Ships (MARPOL) [3,4]. Marine vessels operate with two types of fuel oil: (1) distillate fuel including marine diesel oil (MDO) and marine gas oil (MGO) and (2) residual fuel such as IFO variants and HFO, all fuels that are derived from crude oil [5,6].

The degradation rate of hydrocarbon derivatives in the Arctic sea ice, marine, and soil are not feasible enough due to the cold temperatures combined with the perilous environment, reduced solubility, and high viscosity of hydrocarbons. An emergency mechanical technique was implemented to clean up almost 11 million gallons of oil from the infamous Exxon Valdez tanker oil spill, which displayed some drawbacks. The incident left many habitats and wildlife species still recovering and on the verge of being endangered [7,8]. Physical and chemical clean-up of oil spills is complex and expensive, as it is dependent on the water temperature, composition of oil, amount of oil spilled, and other environmental factors. Therefore, biological treatments such as biodegradation of hydrocarbon pollutants by indigenous microorganisms could exploit the contaminants by using them as a source of energy for growth and other metabolic processes. This method has shown promising results as a potential treatment to eradicate hydrocarbon pollutants [9,10].

The bibliometric approach analyzes and evaluates research publications by monitoring the trajectory and development of the literature attributes in a scientific field. Interpreting and mapping a visualization network of research articles, helps to evaluate the productivity of authors, research groups, journals, institutions, countries, and determine the prospective state of knowledge of the field of study [11,12]. Thus, a comparative bibliometric analysis was employed to analyze the current research trends on the biodegradation of hydrocarbons by the microbial communities in the Arctic and Antarctic marine environments. To the best of the authors’ knowledge, no published study has exclusively explored the biodegradation of hydrocarbons by indigenous microbial communities in both the Arctic and Antarctic marine environments by using a bibliometric framework.

Additionally, a literature analysis was carried out to present the various aspects of microbial degradation of hydrocarbons, particularly in the Arctic marine region, since this was hardly reported compared to the Antarctic. The literature analysis compiled the consequences of hydrocarbon pollution in the Arctic marine environment, followed by the application of diverse microbial communities to degrade hydrocarbons, the metabolic pathway of the oil-degrading microbial community at enzymatic levels, microbial degradation capacity at the metagenomic level, and future prospects of bioremediation in the Arctic marine environment.

## 2. Application of Microbial Community in Biodegradation of Hydrocarbons in Arctic and Antarctic Marine Environments: Comparative Bibliometric Analysis and Literature Review

Comprehensive data mining and analysis of scientific publications for a literature review are the foundation to establish and solve a problem in every aspect of research. A researcher should be able to grasp and understand the overview of a particular field of research. Bibliometrics is a statistical approach used to analyze scientific research or review articles, books, and other publications, which is often referred to as scientometrics [13,14]. This quantitative method mainly entails research assessment in various disciplines by ranking the number of publications according to the authors, source journals, and institutions [15]. The process of exploring scientific information using available information technology (IT) tools or software assists in the analysis and summarization of the gathered data [16].

The present emerging trends in hydrocarbon pollution by microbial communities in cold marine environments are important to evaluate the advancement and dynamics in bioremediation research. This section provides a bibliometrics analysis on the existing literature published during the last 10-year period (from 2009 to 2019). A comparative analysis was performed to highlight the application of microbial communities native to the Arctic and Antarctic cold marine environments in degrading hydrocarbon contaminants. A comparison between both polar regions provides better insights and overview of the proposed study.

The analysis and evaluation of current trends were employed through a data searching of the “Scopus” database produced by Elsevier [17]. This powerful database is reliable and provides substantial information on indexed journals, publications, and a large collection of abstracts and citations encompassing a broad range of subjects [18]. It offers integrated, efficient analytical methods for data extraction, compilation, and content exporting in various formats, which provides an inclusive statistical data of the entire quantity of the world’s research outcomes in the fields of science, medicine, technology, arts, humanities, and social sciences. The premise behind using the Scopus database is that relative to other such databases, at over 23,000 peer-reviewed indexed journals, it has a substantially greater abstract and citation range in interdisciplinary scientific data [19]. In this review, we restricted our assessment to journal articles only, limiting the influence of redundant publications and mitigating false-positive outcomes. Reviews, book chapters, and conference sessions were not taken into consideration, because they contain works that may have been published in multiple sources more than once [13,20].

The process of document collection in relevance to the hydrocarbon degradation by Arctic and Antarctic marine microbial communities as the main theme is represented in the flowchart (Figure 1). The keywords were used as the query string in the Scopus database and analyzed through Microsoft Excel and VOS viewer software (version 1.6.15, Center for Science and Technology Studies, Leiden University, The Netherlands) [21]. A total of 175 and 144 documents pertaining to the Arctic and Antarctic marine environments, respectively, were obtained after search refinement. The occurrence of keywords, publications in countries or territories, affiliations, most productive journals, authors, and most cited documents were analyzed and organized in various visualization representations.

### 2.1. Keyword Co-Occurrence Analysis

Analysis of co-occurrence of keywords is a bibliometric technique that determines the association between topics in the research field of interest. The VOS viewer software constructed visualizing bibliometric maps based on the bibliographical, citation, and all keywords information of 175 and 144 articles that were exported separately. This tool is useful in finding the relatedness or connections between different items or pairs that exhibit specific link strength. The larger the value (numerical) of a link, the greater the strength of a link [22]. Figure 2 illustrates the keywords used and the frequency of occurrence represented by the size of nodes that explains high frequency keywords. Nodes of the same colour are part of a cluster. The closer the distance between each node, the stronger the relatedness of the keywords. A total of five clusters were identified for both research topics by forming groups that correlated with one another’s keywords.

Figure 2a is a bibliometric map of network between the occurrence of keywords in relation to the microbial community that degrades hydrocarbons in the Arctic marine environment. Among the 175 retrieved articles from the Scopus database, 246 keywords were identified that met with a minimum of five occurrences of threshold for mapping. This was presented in different clusters displaying a strong link strength of 49,198. The most frequent keywords indicating the primary clusters were “bioremediation” (green), “microbial community” (red), “hydrocarbons” (purple), “petroleum” (blue), and “microbiology” (yellow). The term “bioremediation” (green) was located in the core of the network and had the greatest interest of this topic, corresponding with the other keywords such as “biodegradation”, “bacterium”, and ‘crude oil’. The main clusters (blue and red), led by “petroleum” and “microbial community”, respectively, have established a high relevance with the “bioremediation” (green) keyword. This suggests that the study of microbial community or bacterium has been exploited in remediating oil pollution in the Arctic sea water. Thus, the occurrence of keywords such as “biodegradation” (95 occurrences, 2424 links), “bioremediation” (94 occurrences, 2498 links), “hydrocarbons” (70 occurrences. 2051 links), “microbial community” (67 occurrences, 1951 links), “marine” (20 occurrences, 527 links), “Arctic” (21 occurrences, 560 links), “Arctic Ocean” (14 occurrences, 345 links) or “Arctic regions” (13 occurrences, 405 links) were highly relevant with other keywords in relatedness to the topic. In comparison, the keywords based on the hydrocarbon degradation by microbial community in the Antarctic marine environment showed 151 keywords that occurred together with 21,229 link strength from a total of 144 publications deposited in Scopus. The most frequently occurred and relevant keywords were selected for network visualization of the keywords co-occurrence in relation to the topic. The bibliometric map in Figure 2b indicates the term “bioremediation” (76 occurrences, 1755 links) occurred the most, followed by other keywords such as “biodegradation” (74 occurrences, 1539 links), hydrocarbon (44 occurrences, 1105 links), “microbial community” (40 occurrences, 1047 links), “marine environment” (12 occurrences, 323 links), and “Antarctica” (five occurrences, 96 links). The main clusters can be depicted by frequently used keywords such as, “biodegradation” (red), “bioremediation” (yellow), “hydrocarbons” (purple), “microbial community” (blue), and “sediments” (green). Our findings elucidated that the terms “bioremediation” and “biodegradation” were often associated with “oil spill”, “bacteria”, and “crude oil”.

The findings showed that the term “bioremediation” was most frequently encountered and was established as the central theme of this topic. The keywords including “microbial community”, “biodegradation”, and “hydrocarbons” were profoundly associated with “bioremediation”. The use of microorganisms to remove or degrade environmental pollutants in oil spill incidents that threaten the marine ecosystem defines bioremediation [23,24]. Biological clean up through microbial biodegradation of hydrocarbons is a promising bioremediation technique, which takes advantage of its own natural mechanisms and is less intrusive and more cost efficient, especially in cold regions [25]. Bioremediation in ice-covered sea and soil during winter is more challenging than in the summer, while most oil drilling and transportation activities are concentrated in the ice-free season. As the temperature rises in summer, oil droplets floating on the surface of the sea are easily attached to the concentrated ice floes and spread even further [26]. The consequences of oil spills in marine environments are further discussed in Section 3.

### 2.2. Trends of Publications in the Last 10 Years

The pattern in the number of publications published on the topic of oil spill remediation or hydrocarbon degradation in cold marine environments with the application of indigenous microbial communities has been extensively increasing from 2009 to 2019 (Figure 3). Many research groups worldwide are conducting extensive research on bioremediation or biodegradation of oil pollutants in low temperature climate regions. Microbial community applications have been found to be less invasive to treat pollution and in addressing the environmental concerns in the Arctic and Antarctic regions. Based on the articles published from the Scopus database, it was elucidated that the number of publications related to the Antarctic environment (107 cumulative publications) was higher than that in Arctic (81 cumulative publications) between 2009 to 2014. However, more articles were published on topics focusing on the Arctic (40 and 58 publications) compared Antarctic (34 and 44 publications) in 2018 and 2019, respectively. This shows that research on the bioremediation of hydrocarbons in the Antarctic and Arctic marine environments is evolving progressively within the last years five years. The current state of advancement and development in technology has caused growing concerns on the environmental impacts to the cold marine ecosystem resulting from the large number of anthropogenic activities.

### 2.3. Analysis of Publications in Subject Areas

The published journal articles in relevance to hydrocarbon degradation by microbial community in cold marine Arctic and Antarctic environments were classified in various subject areas. This highlighted three main areas of knowledge (Figure 4 and Figure 4b), which included environmental sciences, encompassing 36.33% (Arctic) and 26% (Antarctic); immunology and microbiology, encompassing 20.5% (Arctic) and 18% (Antarctica); followed by agricultural and biological sciences, with similar number of publications (17.3% Arctic and 17% Antarctic) in both research areas. Most of the journal articles were attentive in identifying oil-degrading indigenous microorganisms that utilize their metabolic capacity at the genomic and enzymatic levels to naturally degrade hydrocarbon pollutants [27]. Therefore, a large number of papers have been published in the field of environmental science throughout the last decade.

### 2.4. Global Output in Publications and Most Progressive Affiliations

Bibliometric parameters such as countries and affiliated institutions provide opportunities for an international level of collaborations in the particular research field. This also helps some researchers to pursue their research career in established institutions or countries related to their specialization. The trends of publications were analyzed over the last decade based on the 144 and 175 articles related to hydrocarbon pollution in the Antarctic and Arctic marine environments and are presented in a global map (Figure 5). The most productive countries in publishing journal articles were the United States (28% and 23%), China (13% and 16%), and Canada (12% and 11%) for research related to the Arctic (Figure 5a) and Antarctic (Figure 5b) environments, respectively. These countries are the key players in pursuing research in hydrocarbon biodegradation compared to other countries or territories, with their first-rate advancements in technology and scientific research contributing to high publication numbers.

In addition, the most progressive research affiliated institutions in publishing journal articles are listed in Table 1. Based on the Scopus database, the following institutions, namely the Chinese Academy of Sciences in China (4.6%), SINTEF Ocean in Norway (4.6%), and Lawrence Berkeley National Laboratory in the United States (4%) frequently contributed to the publishing of journal articles on this research topic for the Arctic region. Meanwhile, the institutions that have a high impact on this area were Istituto per l’Ambiente Marino Costiero del Consiglio Nazionale delle Ricerche IAMC-CNR (5.5%) and Consiglio Nazionale delle Ricerche (5.5%) in Italy, and Chung-Ang University in South Korea (4.2%). The Chinese Academy of Sciences published a total of 752,710 journal articles by affiliations, which has established a gap with other leading countries. Institutions in Canada, Norway, and Italy also participated actively in this research area to continuously address the oil pollution issues that threaten the environment.

### 2.5. Source Journals Trends, Most Cited Articles Based on Journal Ranks, and Relevance to the Field

The trends of preferred journals from 2009 to 2019 were analyzed based on their number of publications on the topic of hydrocarbon biodegradation through the application of microbial communities in the Arctic and Antarctic marine environments (Figure 6). Figure 6a illustrates the growth of publications in the top 10 journals from 2012 to 2019, with the journal of Marine Pollution Bulletin (8.57%) performing well from 2014 to 2019, producing a total of 15 articles, followed by 12 articles from the journal Frontiers in Microbiology (6.86%), and Applied and Environmental Microbiology (2.86%), with five documents published. The extracted articles from these top 10 source journals were further analyzed to identify the most cited documents related to this research area (Table 2). The citations of a document reflect the quality of the document published [28]. The most cited article with total citations of 105 that was ranked seventh on the list of productive journals was published in 2015 by the International Biodeterioration and Biodegradation journal. At least 80% of the 10 journals demonstrated an emerging growth in published articles within the last five years. Most of the articles were published by Elsevier (50% in Table 2) and Springer, with both equally publishing similar numbers of articles as shown in Table 3. High impact academic journal publishers such as Elsevier and Springer provide large scale of bibliometric electronic databases across various disciplines [28].

The literature search from the database of 144 documents showed an extensive publishing by the top 10 journals in the study of hydrocarbon degradation by microbial communities from the Antarctic marine environment. According to Figure 6b, the pattern in number of documents published annually from 2009 to 2019 progressively increased, though with some decline in certain years. By comparing with Figure 6a, the journal Frontiers in Microbiology (8.33%) was ranked first with a total publication of 12 articles, while the journal Marine Pollution Bulletin published 11 articles (7.64%), followed by the journal International Biodeterioration and Biodegradation (4.86%), with seven documents.

In addition to this, the Cite Score (Elsevier-Scopus), Clarivate Analytics of Journal Citation Reports (JCR), and Impact Factor (IF) information were evaluated based on the statistics of number and quality of documents and source journals. According to the Cite Score in 2019, nine journals (Table 2) and eight journals (Table 3) had Cite Scores of above five, and there were two journals that scored above 10, which were the Environmental Science and Technology (12.6) and Journal of Hazardous Materials (13.1). This information allowed authors to select suitable journals to add up the significance and novelty of their research work. The authors should also consider various metrics in measuring the journal impact and compare them with other published articles in the similar research field [29].

### 2.6. Most Prolific Authors

Leading authors in publishing relevant articles were analyzed from the Scopus database based on the bibliometric parameters including author (Scopus) ID, current affiliations, highest cited document, total citations and publications, h-index, journal, and cited by. The total number of publications (175 and 144 journal articles) on the study of hydrocarbon degradation by microbial community in the Arctic and Antarctic marine environments are summarized in Table 4 and Table 5. These prolific authors produced a high number of articles and achieved high scientific impact in research. The h-index is an important metric used to evaluate the cumulative impact of a researcher’s publication output and performance by comparing the publications to citations [29,30].

Based on Table 4, out of a total of 175 publications, the three most prolific authors who were actively publishing articles related to the study of hydrocarbon degradation in the Arctic marine environment by utilizing native microbial communities were Cappello, S. (5.1%), Jeon, C.O. (4%), and Madsen, E.L. (2.9%). Meanwhile, out of a total of 144 documents retrieved from the database, the top three productive authors who published articles from 2009 to 2019 (Table 5) for the Antarctic environments were Brakstad, O.G. (4.2%), Netzer, R. (3.5%), and Anderson, J.A. (2.8%). Overall, Andersen, G. L., who published an article that was cited 133 times, had the highest h-index (61) and citations per publications (CPP) of 191.43 compared to other authors, whereas Jeon, C.O., had a CPP of 23.21 and a h-index of 49, the highest in comparison to authors listed in Table 5. Moreover, the most productive author in this research study focusing on the Arctic marine environment was Brakstad, O.G, with a h-index of 22, CPP of 28.56, and the author’s highest cited publication was cited by 31 other published articles. For the Antarctic environments, Cappello, S., was the most prolific author, with a h-index of 25 and CPP of 26.22 and cited by 62 other published articles. He is affiliated with the Consiglio Nazionale delle Ricerche in Rome, Italy, whereas Brakstad, O. G, along with Netzer, R, and Ribicic, D., are affiliated with the SINTEF Ocean in Trondheim, Norway. Both of their affiliations were ranked second in terms of their publications number in last 10 years (Table 1). Three (Almeida, A. P., Cleary, D.F.R., and Cunha, A.) out of the 10 authors have affiliations with the Universidade de Aveiro, Aveiro in Portugal (ranked fifth). This indicated that even though China and Italy (Table 1) produced high quantities of publications, more active authors were linked with institutions in Norway and Portugal that have great interest in investigating the applications of microbial communities in hydrocarbon degradation in the Arctic and Antarctic marine environments.

### 2.7. Limitations of Bibliometric Study

This study recognizes several shortcomings that could be explored in further research. The results of the bibliometric analysis are characterized by the research area, the selected timeframe, database, keywords and articles analysis, and the evaluated variables. Therefore, comparisons with multiple data outputs such as Web of Science and Scopus will be effective to find notable differences to assess the latest research trends [22,31]. In this study, only published articles have been taken into account; thus, publications such as books, proceedings, and other types of documents could be taken into consideration in the future. This paper utilized a comparative bibliometric study between two different focus of studies by quantifying the articles published over the years followed by extracting the information and then presented using network analysis. However, several other methods such as qualitative measurements using questionnaires on the opinions of people with various backgrounds and seeking experts for constructive discussions could be combined with the traditional bibliometric approach (hybrid model) [30,31]. Alternatively, a topology structure and model mapping could also be used to determine the extent of advancement over different fields or disciplines and increase the robustness of the bibliometric analysis [13,32].

## 3. Consequences of Hydrocarbon Pollution in Arctic Environments: Seawater, Marine Sediments, and Coastal Environments

Our bibliometric analysis showed that the research output in the bioremediation or biodegradation of hydrocarbons by Arctic marine microbial communities outnumber those for the Antarctic marine environment. Subsequently, an extensive literature analysis will identify various aspects of microbial degradation of hydrocarbons, from the fate of oil spills in the ocean to the enzymes or genes that are responsible for biodegradation (Section 3, Section 4 and Section 5). Oil containment and elimination in the marine setting, especially in cold conditions, pose challenges in removing pollutants completely from the environment. Although much of the spilled oil would have been physically collected through containment booms, a substantial amount of the residual oil during clean-up will wash up to the coast or ashore, while some resurfaced oil will float on the surface of water, while others were deposited in the marine sediments depending on the oceanic currents (Figure 7) [33,34,35].

In the Arctic, the primary sources of hydrocarbon pollution generally come from offshore drilling, oil tank blowouts or leaks, fuel transportation, leakage from ships, pipeline leakages, runoffs from land or river, discharges from water, and long-range transportation of pollutants via water. The marine ecosystem is largely affected, especially by large-scale oil spill disasters such as the Exxon Valdez tanker oil spill incident in Alaska (1989), Deepwater Horizon oil rig explosion in the Gulf of Mexico (2010), and Norilsk diesel fuel spill in Russia (2020). Such disasters were caused by human errors, illegal disposal of oil waste, and bunkering of oil among the ships [37]. There were other oil spill incidents, from low to high level of discharge of oil, that have been recorded as listed in Table 6.

### 3.1. Seawater of the Marine Environment

The oil hydrocarbons spilled on marine surfaces encounter several weathering processes including photo-oxidation via carbon dioxide and water, emulsification, dissolution dispersion, evaporation, sedimentation, adsorption, as well as microbial biodegradation [44,45]. Oil spills in the marine environment occurs mainly due to ship accidents during the transportation of crude oil, severe or extreme oceanic currents and weather conditions, risky geographical regions (prone to sea ice melting), and possible human errors. The spilled oil could spread further, forming slicks over the surface of the Arctic seawater. Approximately, 5 × 10^6^ m^2^ of slick can be formed from 1000 kg of oil spilled, which depletes the oxygen level and causes pH changes in the seawater due to the disruption of gaseous (oxygen and carbon dioxide) exchange [46,47].

### 3.2. Marine Sediment

Accidental or unnecessary waste discharges associated with offshore drilling such as production water may be deposited in the sediments of the marine environment. Moreover, to sustain the pressure and rushing of the hydrocarbons into productions wells, tonnes of injection water are introduced into the system. Despite clean-up efforts that are based on regulatory guidelines, some of the low molecular weight hydrocarbons, inorganic salts, additives, monocyclic aromatic hydrocarbon (BREX), polycyclic aromatic hydrocarbons (PAHs), and other aromatic compounds remained in the environment [47]. Apart from operational waste discharges, huge volumes of natural gas and crude oil leak out of the ocean’s seabed and sedimentary rock, especially in areas rich with oil resources including in the sandstones of Alaska, Timan-Pechora basin in Russia, and submarine seeps in Nuussuaq peninsula of Greenland. Oil seeping from the seabed acts in a similar manner to oil drilling or spills by forming massive slicks that circulate through long-range atmospheric and ocean currents [48].

### 3.3. Coastal or Shoreline of Arctic Ocean

The integrity of the Arctic’s ecosystem, including marine mammals, birds, corals, fish, and wildlife at coastal and shore regions have been destroyed in the pursuit of natural gas and crude oil exploration. The wildlife, marine animals, and birds manifested behavioral changes in their motility, feeding, and physical activities and were affected by disrupted growth and reproduction. Most of the hydrocarbons are less toxic when present at low amounts, although some aldehydes with low molecular weight, (monocyclic aromatic hydrocarbon) MAHs, and (polycyclic aromatic hydrocarbon) PAHs are stable, extremely carcinogenic, and toxic [49]. The clean-up and recovering process are extremely challenging for ice-infested waters in the Arctic and certainly cause diverse negative impacts to the overall ecology. Most carcinogenic hydrocarbons are classified as high molecular weight aromatic compounds, which pose high potential risk and increased toxicity. Among the composition of fuel hydrocarbons, PAHs have higher chances for bioconcentration due to their low volatility, insolubility in aqueous system, and lower rate of degradation. An early study suggested that the toxicity and persistence of PAHs in the environment are mainly caused by their large size due to their more fused benzene ring. Derivates of lighter hydrocarbons are hydrophilic and soluble up to 2000 ppm [50]. Wildlife populations in the Arctic are being chemically exposed to the toxic hydrocarbons and currently threatened at an alarming rate, and thus the potential biological effects need to be assessed. The toxicological effects of hydrocarbons on the marine ecosystem, coastal wildlife, flora, and humans require more attention by researchers.

### 3.4. Toxicological Effects of Hydrocarbon Pollution on Different Marine Environments.

Various fish are affected by PAHs and their derivatives, including naphthalene that was observed to reduce the larval growth of fathead minnow (*Pimephales promelas*) [51]; induced genotoxic damages in eel (*Anguilla anguilla*) [52] and Arctic copepods (*Calanus finmarchicus* and *C. glacialis*); damaged deoxyribonucleic acid (DNA) and inhibition in acetylcholine esterase (AChE) observed in Littoral crab (*Carcinus aestuarii*) [53], Icelandic scallop (*Chlamys islandica*), blue mussel (*Mytilus edulis*), and red king crab (*Paralithodes camtschaticus*) that led to nervous system damage, behavioral changes in swimming to escape, and induced oxidative stress [54,55,56].

In marine mammals, PAHs induces oxidative enzyme activity in beluga whales (*Delphinapterus leucas*) [57] and an orca pod (AT 1) [58] was yet to recover from the Exxon Valdez oil spill from 25 years ago. The Arctic coastal and shoreline regions are occupied by a large and diverse number of sea- and shorebirds, which are the most vulnerable and sensitive species to oil pollution due to regular contact with water surfaces. Floating oil covers the thick fur or feathers of polar bears, sea otters, seals, walruses, and seabirds, which prevents them from staying afloat and reducing insulation, leading to hypothermia, drowning, smothering, and toxic hydrocarbons ingestion through the food web [58,59]. Meanwhile, toxic fumes inhalation increased the mortality in harbor seals through stress, brain lesions, and disorientations [8].

Oil spills or pipeline leakages absorbed by the soil at the shorelines resulted in reduced soil fertility that inhibits the growth of some plants. However, oil spreads minimally in soil, as hydrocarbons transform through blending in soil or snow ice, photo-oxidation, and microbial degradation [58]. Plant life in the Arctic suffers much higher consequences compared to other terrestrial or tropical regions. The hydrocarbons from oil spills can severely damage plants due to the vulnerable thin surface cover [8]. Moreover, the Arctic vegetation requires a longer time to recover from exposure to the toxic compounds due to the low temperature and shortage of nutrients. Exposure and potential health effects of PAHs are also of concern in humans due to their chronic effects such as reduced immune system, damage to the liver and kidneys, cataracts, and respiratory abnormalities [60]. General routes of the carcinogens’ exposure are through nasal inhalation, ingestion, and skin contact via oil spills in soil, water, or air. Workers especially at oil drills, transportation, or spill sites suffer from acute symptoms, including vomiting, skin inflammation, and eye irritation [61,62]. Aromatic hydrocarbons are the main contributors of carcinogenicity in humans and animals, which has been made evident through cellular damage via mutations that lead to cancers of the gastrointestinal, lung, bladder, as well as skin due to long term exposures [63].

## 4. Application of Microbial Community in Biodegradation of Hydrocarbons in Arctic Environments

This section discusses the applications of microbial community that focus mainly on the Arctic environments in degrading or remediating oil pollutants. The treatment of hydrocarbon contaminants, a combination of both mechanical and biological techniques or either one, are used to contain and remediate oil spills in the Arctic soil and ocean. The remoteness of the Arctic environment and its unpredictable conditions make oil spills extremely dangerous to animal habitats. Indigenous microbial floras concentrated in the seabed close to natural seeps have been found to degrade huge volumes of oil spills from the Deepwater Horizon and Exxon Valdez incidents at low temperatures and as deep as 1000 m [64]. Physical booms, skimmers, and chemical dispersant that are used in the initial response to clean up oil spills are expensive, as well as requiring more manpower, prolonged response, and ecological side effects. However, some mechanical and chemical clean-up methods are not able to remove the hydrocarbons once the oil is emulsified in water or soil surfaces, which further increases the damage and recovery period of the ecosystem [65,66,67].

The nature of hydrocarbon compounds is not viable for degradation, especially in low temperatures. Therefore, the factors including temperature, pH, nutrients, and oxygen can influence the microbial activity [68]. Psychrophilic and psychrotolerant bacteria that are derived from cold environments have increased potentials in the decontamination of hydrocarbons from oil-polluted sites in the Arctic. The minimum, optimum, and maximum temperatures in assisting growth are differentiated as follows: for psychrotolerant bacteria, temperatures of <0−5 °C, <15 °C, and <20 °C are required; whereas for psychrophile bacteria, <0 °C, <15 °C, and <20 °C are required [69]. Reduced enzymatic activity is associated with lower degradation rate at low temperatures [65,70]. The maximum level of hydrocarbon metabolism can be achieved at temperatures between 30 to 40 °C. In the Arctic, the growth of hydrocarbon-degrading Proteobacteria, including *Psychrobacter*, *Pseudoalteromonas,* and *Pseudomonas* spp. [71,72] are promoted through the addition of hydrocarbons. The marine bacteria, *Marinobacter* [73], exhibited degradation capacities of 20–50% at temperatures as low as 1 °C and almost 80% hydrocarbon degradation for *Oleispira* at 4 °C [74].

Biostimulation, which is the incorporating of nutrients for hydrocarbon degradation, stimulates the nitrogen content for assimilation and induces growth. The oil-contaminated soils in the Arctic have low levels of nitrogen concentrations due to an increase in carbon, which limits the bioremediation process. The carbon sources essential for degradation are supplemented through hydrocarbon and water supplying oxygen and hydrogen [75,76]. Thus, the addition of nutrients is dependent on the mechanisms of the specific bacteria to uptake and promote assimilation for rapid mineralization of hydrocarbons. The effectiveness of PAHs degradation has been observed in diversely growing *Sphingomonas* strains that can adapt to various nitrogen concentrations [77,78].

The addition of hydrocarbon-degrading bacteria, algae, fungi, algae, or yeast to prevailing population of microbes is known as bioaugmentation, which is inexpensive and allows in-situ bioremediation. The oil-degrading microorganisms isolated from oil spill sites increase the degradation capacity of organic pollutants in conjunction with indigenous microbiota, with high tolerance towards the toxicity of hydrocarbons [79,80]. The presence of various Arctic microorganisms has been observed in crude and diesel oil through stimulation of hydrocarbons that degrade aromatic compounds (Table 7) [81,82,83]. In cold regions, hydrocarbonoclastic bacteria are important in hydrocarbon degradation at low temperatures [84,85,86]. Furthermore, other microbes such as the fungi *Phialophora* spp. and *Hormoconis resinae* [87,88] are known to degrade hydrocarbon pollutants in Antarctica. Filamentous fungi and yeast in terrestrial environments can degrade PAHs in the presence of responsible genes for degradation. Experimental research on hydrocarbon-degrading fungi and algae is yet to be thoroughly investigated in the cold regions of the Arctic, but some studies have reported on the ability of fungi and algae to degrade aromatic compounds and other derivatives of hydrocarbon in the Antarctic and terrestrial regions [89,90,91].

## 5. Metabolic Pathways of Oil-Degrading Microbial Community at the Enzymatic Level

Complex hydrocarbon compounds in nature degrade according to the following order of biodegradability (least to most): linear alkanes—branched alkanes—low molecular weight alkyl aromatics—monocyclic aromatic hydrocarbons (MAHs)—cyclic alkanes—polycyclic aromatic hydrocarbons (PAHs)—high molecular weight asphaltenes [94,95]. Aromatic hydrocarbons are the least biodegradable in comparison to alkanes; therefore, the physico-chemical properties, availability of substrates, and environmental conditions are necessary to eliminate them through eco-friendly and feasible bioremediation techniques. Previous studies largely discussed and reported on the most toxic contaminant, namely, PAH biodegradation, due to its persistency and ubiquitous nature towards the marine ecosystem [96,97].

The petrogenic origins of PAHs were identified in the Arctic sea through an extensive sampling with depths of 367–4320 m, while some low molecular weight PAHs were abundant in the Arctic Ocean North–South Transect, notably close to the land [98]. Moreover, Yunker et al. [98] also revealed deep sea sediment microbial members, namely, *Pseudomonas*, *Dietzia*, *Pseudoalteromonas*, *Halomonas*, *Cycloclasticus,* and *Marinomonas* found in the Arctic Ocean, which were responsible for PAH degradation. Cold-acclimated microorganisms such as PAHs-degrading *Pseudomonads* (*β*-Proteobacteria) and *Sphingobium* sp. C100 (*α*-Proteobacteria) were enriched in the deep-sea sediments of the Arctic Ocean [99], *Marinomonas profundimaris* D104 [100], while *Colwellia* was present in the Svalbard sea ice of the Arctic [101,102]. The majority of hydrocarbons (94%) were degraded by the bacterial communities present in sub-ice seawater of the Arctic by removing almost all the biodegradable alkanes and similarly for the methylated forms of PAHs. It was identified that anaerobic microbial communities of *Gammaproteobacteria* and *Bacteroidetes* are the predominant groups underneath the shore-fast ice of the Arctic sub-ice, while *Alphaproteobacteria* majorly dominates the surface of seawaters [103].

Hydrocarbon degradation involves two common pathways: aerobic (oxygen-dependent) and anaerobic (oxygen-independent) (Figure 8). Microbial degradation of hydrocarbons under aerobic conditions involves an oxidative process performed by oxygenase and peroxidase as an initial attack. This pathway is often necessary in hydrocarbon degradation, as the amount of oxygen is abundant on the surface of seawaters, in contrast with the low availability of oxygen in the water columns and deep seawater [92,104]. The bottom of the ocean bed is anoxic where the oxygen concentrations are limited, depending on the amount of oil and level of oxygen renewal by the ocean current [105]. A previous study reported that the enzymatic attack is highly influenced by the composition of hydrocarbons in oil, as the linear and some branched alkanes are degradable, unlike aromatic compounds [106].

Analysis of potential microbial community of PAHs degraders found that they utilized genes encoding for the PAH monooxygenase or dioxygenase (large alpha subunit), which could be utilized as biomarkers [107]. These enzymes are members of the ring-hydroxylating oxygenase (RHOs) or ring-hydroxylating dioxygenase (RHDs), a large enzyme family that activates the compounds to generate a cis-dihydriol from the dihydroxylation of one of the aromatic rings [97,108]. Next, the cis-dihydriol dehydrogenase rearomatizes the aromatic ring of hydrocarbons, forming di-hydroxylated intermediates that are oxidised into catechol, which is the primary intermediate of oxygen-dependent degradation [109]. The metabolic advantage of aerobic catabolism of hydrocarbons is that it is more rapid due to the presence of oxygen (O_2_) molecule as an electron acceptor through the introduction of one or two atoms of O_2_ to a substrate (hydrocarbon) [110].

Apart from that, anaerobic biodegradation of hydrocarbons requires compounds other than oxygen that can act as electron acceptors such as sulphate, nitrate, and ferric chloride [111]. A proposed general pathway includes two primary activation mechanisms, namely direct carboxylation of aromatic rings by carboxylase originated from UbiD-like proteins or a methylation pathway that methylates aromatic compounds through methyltransferase [112]. The second mechanism comprises the supplementation to fumarate by the catalysis of a glycyl-radical enzyme identified as naphthyl-2-methyl-succinate synthase [113,114]. Both mechanisms advance with activation of coenzyme A via the process of β-oxidation [115]. Enzymatic pathway activation requires some time to be fully functional for hydrocarbon degradation [116].

Microbial communities possessing high enzymatic capacity can degrade complex hydrocarbons through transformation of contaminants into less toxic forms. The distinctive catalytic properties evolved in the psychrophilic microbial community express cold-adapted enzymes that have enhanced flexibility, reduced thermal stability, and greater catalytic effectiveness in extremely cold environments [118,119]. These enzymes are expressed by genes adapted to the low temperature conditions, which improve the catalytic activity of hydrocarbon degradation through certain environmental adaptation processes [120]. The microorganisms enhance the fluidity of their cell membrane and increase the short-chain fatty acids and unsaturated fatty acids at low temperatures. Psychrophilic enzymes have an elevated specific activity compared to that in mesophilic and thermophilic microorganisms [121,122]. Thus, further investigations are necessary to discover more cold-adapted enzymes capable of degrading hydrocarbons from the polluted environments in the Arctic regions.

## 6. Microbial Degradation Capacity at the Metagenomic Level

The rate and magnitude of hydrocarbon biodegradation in cold marine environments are considered to be crucial elements and require understanding in the removal of compounds by microbes. Multiple aspects such as pollutants, oxygen, pH, nutrient, temperature, and the wide range of microbial community have been found in the contaminated regions. Microbial communities present in the low temperatures of Arctic marine ecosystems are capable of degrading hydrocarbons as a carbon source and promoting energy production while utilizing the bioavailability of nutrients such as nitrogen [123]. The molecular analysis of functional genes that are active in oil-degrading microbial communities can provide different insights at the metagenomic and enzymatic levels [27,124]. This method can be used to collect information on the distribution and diverse range of potential hydrocarbon-degrading microbial communities in the Arctic marine environments, including coastal, marine sediments, and seawater. These cold adapted marine microbial communities can act as innovative model organisms for the bioremediation of hydrocarbon pollutants.

Recent advancements in analyzing and understanding the metagenomics of the microbial communities are essential to access the taxonomic and composition of functional genes in the biodegradation of oil pollutants. The study of genetic material of intact communities is known as the metagenomics process. Metagenomics involve the use of next-generation sequencing (NGS) with the latest approaches including whole-metagenome sequencing (WMS) [125,126]. The diversity of hydrocarbon degrading-bacteria present in the sediment of the Arctic Ocean has been determined through a large scale of 16s rRNA analysis in a study that assessed the presence of PAHs. Samples collected from 19 various deep-sea sediments (Chukchi Plateau, Makarov Basin, Canada Basin, and Alpha Ridge) showed members of Halomonas, Cycloclasticus, Pseudomonas, Marinomonas, Dietzia, and Pseudoalteromonas genera that exhibited the capability to mineralize PAHs [92]. In another study, the taxa Colwellia and Oleispira were found to have higher responses to oil in the Arctic seawater. These taxa were associated with the functional genes alkB, nagG, and pchCF in hydrocarbon degradation and dispersant Corexit 9500 [127]. In addition, the obligate hydrocarbonoclastic or heterotrophic genera Pseudomonas, Cobetia, Thalassomonas, Marinobacter, Moritella, Shewanella, and Aestuariicella found in the deep-sea water of the Arctic have also been shown to favor oil degradation [98].

The application of metagenomics in the largest oil spill accident to-date, the 2010 Deepwater Horizon drilling rig explosion, provided an opportunity for researchers to explore and deepen the knowledge of the biodegradation of hydrocarbon pollutants. PAH-degrading bacteria such as Cycloclasticus [128], Marinobacter [129], Pseudoalteromonas, Marinomonas, Halomonas [130], Sphingomonas [131], and Vibrio [132] were identified from coastal sediments through metagenomics [133]. The phdCI gene that encoded for carboxylate isomerase responsible for degrading naphthalene was largely found in oil plume samples. Moreover, the Oceanospirillales, which belong to the Y-Proteobacteria, including Oleispira antarctica, Oleiphilus messinensi, and Thalassolituus oleivorans, have been found in Arctic seawater [118]. The Deepwater Horizon incident left a negative impact on the marine environment due to the large portion of oil remaining in deep-sea sediments, including the deep-water plume of oil. Oleophilic bacterial samples, Colwellia rossensis and Oleispira antarctica from natural North Sea fjord seawater and Arctic seawater, are capable of expressing GyrB genes. Through this molecular approach, the identified hydrocarbon-degrading genes could be utilized as a proxy in detecting oil contamination in the seawater of cold environments [134]. The metagenomics approach can also improve the understanding of the fate of oil spills and the dynamics of microbial structure.

## 7. Future Prospects of Bioremediation in the Arctic: Limitations and Suggestions

Emerging technologies in bioremediation strategies have been introduced with the main purpose to restore contaminated ecosystems in an environmentally sustainable and cost-effective manner. The ecosystem in the extreme Arctic conditions requires a longer time to recover from pollution [135]. Ex-situ bioremediation is widely implemented as the initial response towards accidental oil spills, tanker blowouts, or pipeline leakages to control harmful pollutants and minimize damage to marine and wildlife ecosystems. Bioremediation in the Arctic is governed by time and expense constraints, internal regulations, and guidelines. The time taken for the ecosystem to recover is much longer than temperate environments for the same concentrations of hydrocarbons, for many months or even decades [103]. This is due to the slower rate of natural attenuation and thermal incineration that poses a threat to permafrost and descending pollutant migration [136]. Moreover, based on the literature search and readings, studies on Arctic bacterial degradation of hydrocarbons are prominent; however, less is known about the application of algae, fungi, or other microbial communities. Therefore, with availability of resources and new technologies, it could be feasible to explore more and solve the problem of oil pollution.

Bioaugmentation is one of the in-situ bioremediation techniques that is less common in the Arctic due to limited nutrients levels that inhibit successful bioremediation of hydrocarbon contaminants. Therefore, the addition of nitrogen, carbon, and phosphorous sources is essential to optimize the growth of microbial communities and increase the efficiency of biodegradation of hydrocarbons [137,138]. Biodegradation of hydrocarbons by bacteria [139] is popular in the Arctic owing to their availability and adaptational capability, but less information is available on the functionality of fungi or algae. Phytoremediation is a potential bioremediation method to replace expensive techniques in cold regions. Plant species in cold regions may assist in soil decontamination. However, it is highly dependent on various growth factors such as water, soil nutrients, temperature, soil chemistry, and seasons [140,141]. Similarly, another primary producer is the sea ice algae population that contributes 3% to 50% of nutrients for copepods, zooplanktons, and krill that are then eaten by secondary consumers, fish, and whales [142]. Apart from that, bioreactor-based bioremediation treatments allow biodegradation of hydrocarbons in an optimal condition. The efficiency of bioreactor-based degradation relies on the ability of microbes to bind to inert packing that produces a high biomass [143]. Therefore, various biotechnological approaches can be used as an alternative over mechanical techniques to ensure the sustainability of the Arctic ecosystem.

## 8. Conclusions

The recent developments in biotechnology and engineering have produced new technologies to deliver novel and effective methods in managing polluted Arctic marine environments. Bibliometric analysis is valuable to identify the gaps of knowledge in the field of study. Polar regions are immensely vulnerable to rising anthropogenic activities that pose risks to the environment because of the high possibility for the next major oil pollution to occur. Bioremediation approaches improve the sustainability of the environment via in-situ remediation, including applications of native cold-acclimated microbial communities that safely degrade harmful hydrocarbons. The investigations on biodegradation of hydrocarbons in cold regions remain limited, and more information through enzymatic and metagenomics insights are required to respond and prepare towards the consequences of oil pollution in the Arctic marine ecosystem. Over the years, studies had proven that biodegradation of hydrocarbon via indigenous microbial communities is able to enhance the remediation process. This review has provided an overview for further scientific research and highlighted future research that are needed for microbial applications in the bioremediation of oil pollutants.

## Figures and Tables

**Figure 1 ijerph-18-01671-f001:**
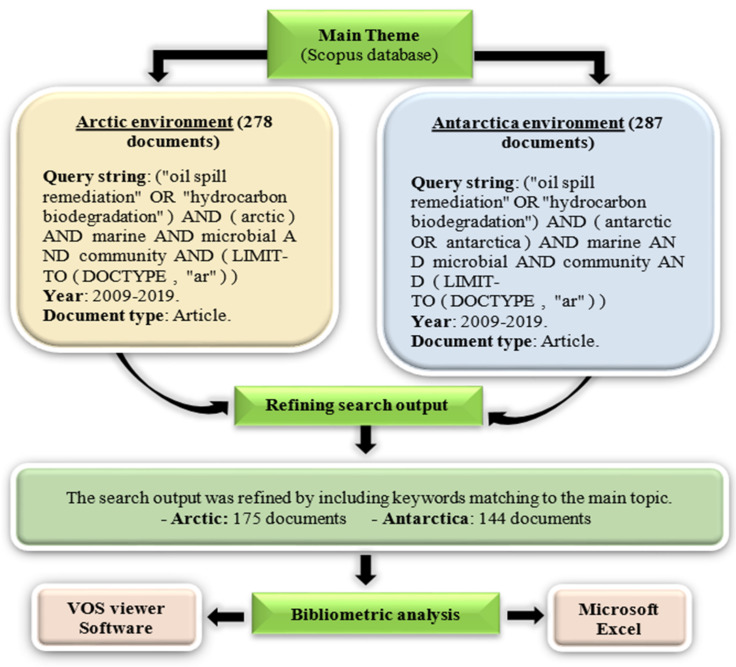
Flowchart of data collection of documents published in relevance to the main theme: hydrocarbon degradation by microbial communities in both Arctic and Antarctic marine environments.

**Figure 2 ijerph-18-01671-f002:**
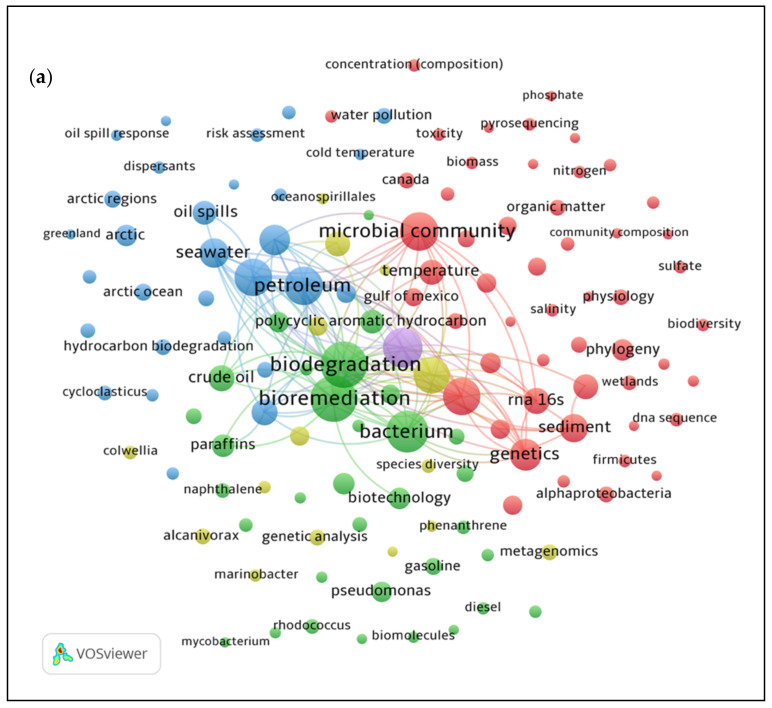
Keywords occurrence on hydrocarbon degradation by (**a**) Arctic and (**b**) Antarctic marine microbial community.

**Figure 3 ijerph-18-01671-f003:**
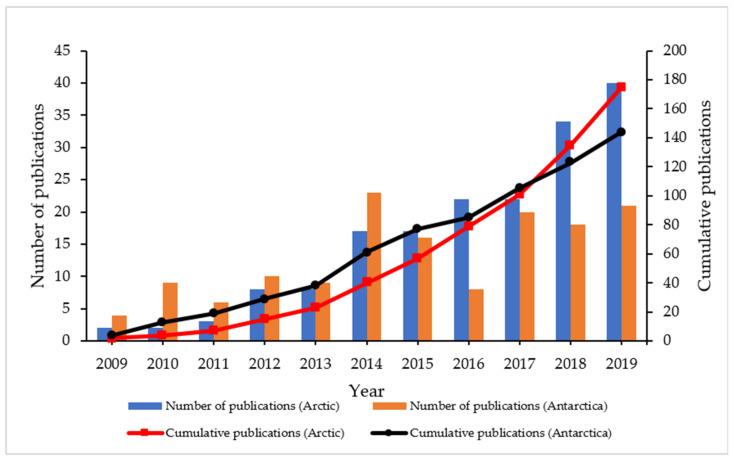
Number of publications from 2009 till 2019 related to hydrocarbon degradation by microbial communities in Arctic and Antarctic marine environments.

**Figure 4 ijerph-18-01671-f004:**
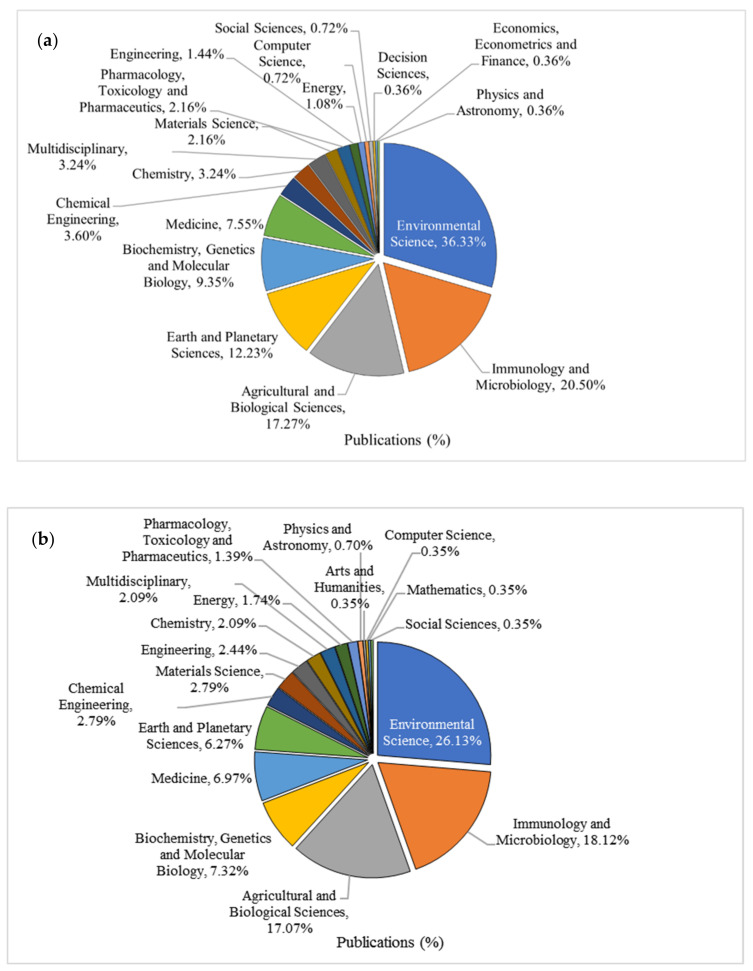
Distribution of publications by various subject areas based on the hydrocarbon degradation by microbial community in marine environments of (**a**) Arctic and (**b**) Antarctica.

**Figure 5 ijerph-18-01671-f005:**
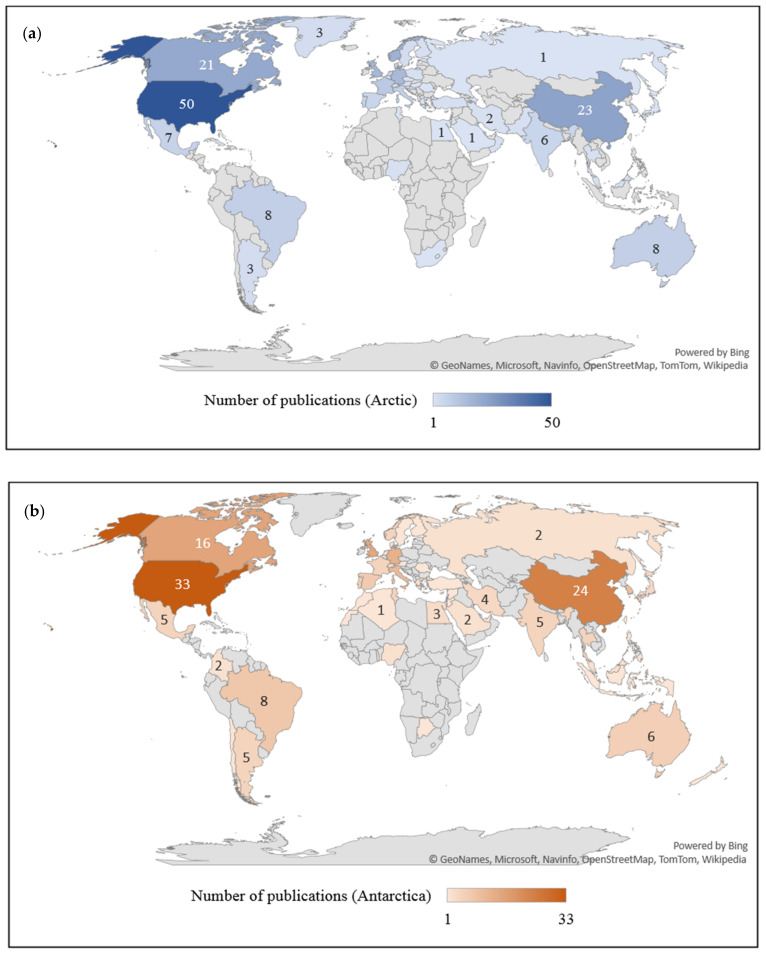
Number of journal articles published by countries or territories related to hydrocarbon degradation by (**a**) Arctic and (**b**) Antarctic marine microbial community.

**Figure 6 ijerph-18-01671-f006:**
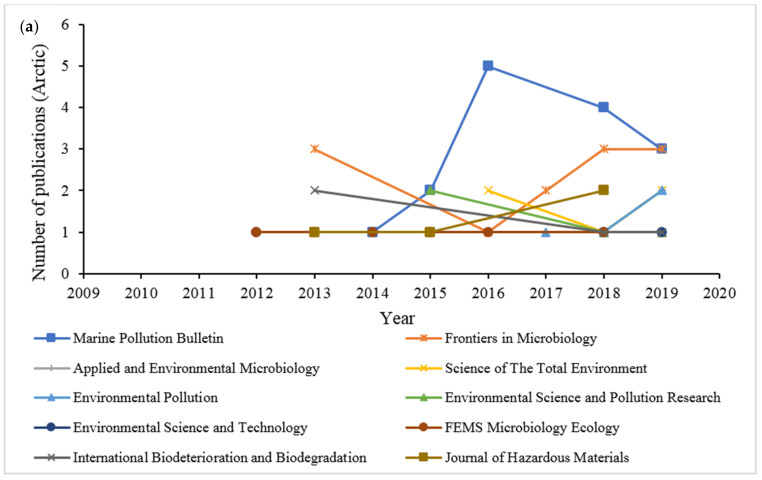
Trends of the top 10 source journals based on the degradation of hydrocarbon by microbial community in (**a**) Arctic and (**b**) Antarctic marine environments between 2009 to 2019.

**Figure 7 ijerph-18-01671-f007:**
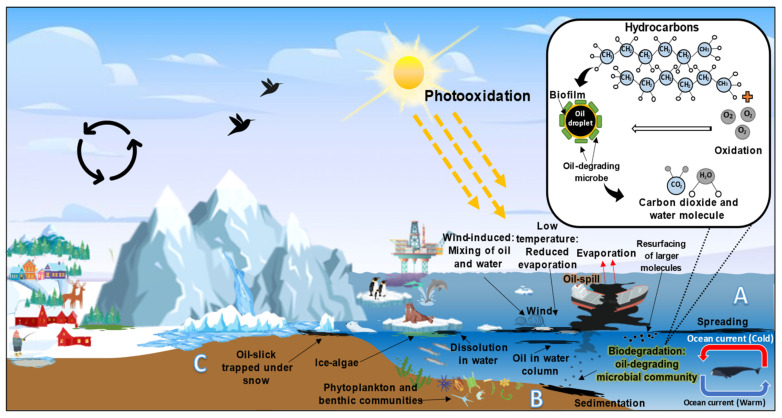
The fate of oil spills into the Arctic marine environment from the aspects of the (**A**) seawater, (**B**) marine sediment, and (**C**) coastal environments (Adapted from: [34,36]).

**Figure 8 ijerph-18-01671-f008:**
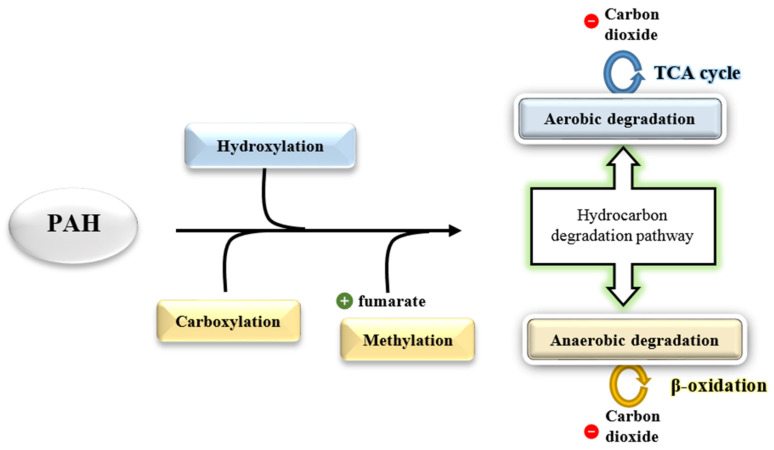
The primary activation mechanism in PAH degradation via the initial step of activation (reconstructed from [34] through adaption from [115,117]).

**Table 1 ijerph-18-01671-t001:** Top ten ranked affiliations with the number of publications between 2009 and 2019 based on the research on degradation of hydrocarbon by Arctic and Antarctic marine environments.

Regions	Rank	Affiliations	Number of Publications	Country	TP by Affiliations
	1	Chinese Academy of Sciences	8	China	752,710
	2	SINTEF Ocean	8	Norway	1892
	3	Lawrence Berkeley National Laboratory	7	United States	87,396
	4	Fisheries and Oceans Canada	7	Canada	13,334
	5	Norges Teknisk-Naturvitenskapelige Universitet	7	Norway	75,976
Artic	6	Consiglio Nazionale delle Ricerche	7	Italy	202,399
	7	Istituto per l’Ambiente Marino Costiero del Consiglio Nazionale delle Ricerche IAMC-CNR	6	Italy	1573
	8	National Research Council Canada	5	Canada	57,233
	9	The University of Tennessee, Knoxvill	5	United States	72,041
	10	University of Aberdeen	5	United Kingdom	55,006
	1	Istituto per l’Ambiente Marino Costiero del Consiglio Nazionale delle Ricerche IAMC-CNR	8	Italy	1573
	2	Consiglio Nazionale delle Ricerche	7	Italy	202,399
	3	Chung-Ang University	6	South Korea	23,620
	4	Chinese Academy of Sciences	6	China	752,710
	5	Universidade de Aveiro	5	Portugal	35,047
Antarctica	6	University of Essex	4	United Kingdom	24,631
	7	Sorbonne Universite	4	France	160,067
	8	Ministry of Education China	4	China	446,329
	9	Consejo Nacional de Investigaciones Científicas y Técnicas	4	Argentina	88,317
	10	Cornell University	4	United States	180,481

**Table 2 ijerph-18-01671-t002:** Top ten most productive journals on degradation of hydrocarbons by Arctic microbial community research with their most cited articles.

Rank	Journal	TP (%)	TP	IF (2019) JCR	TC (2019) JCR	TC (2019) Scopus	Cite Score 2019	The Most Cited Article Based on Source	Year	Cited by	Publisher
1	Marine Pollution Bulletin	8.57%	15	4.049	7000	21,687	6.7	Biodegradation of marine crude oil pollution using a salt-tolerant bacterial consortium isolated from Bohai Bay, China	2016	34	Elsevier Inc.
2	Frontiers in Microbiology	6.86%	12	4.236	24,092	65,387	6.4	The microbial nitrogen cycling potential is impacted by polyaromatic hydrocarbon pollution of marine sediments	2014	43	Frontiers Research Foundation
3	Applied and Environmental Microbiology	2.86%	5	4.016	4761	17,047	7.1	Corexit 9500 enhances oil biodegradation and changes active bacterial community structure of oil enriched microcosms	2017	41	American Society for Microbiology.
4	Science of the Total Environment	2.86%	5	6.551	45,650	133,587	8.6	Biodegradation of dispersed Macondo crude oil by indigenous Gulf of Mexico microbial communities	2016	41	Elsevier B.V.
5	Environmental Pollution	2.29%	4	6.793	17,416	47,665	9.3	Metagenome enrichment approach used for selection of oil-degrading bacteria consortia for drill cutting residue bioremediation	2018	20	Elsevier Ltd.
6	Environmental Science and Pollution Research	2.29%	4	3.056	17,158	53,503	4.9	Dynamics and distribution of bacterial and archaeal communities in oil-contaminated temperate coastal mudflat mesocosms	2015	23	Springer Verlag
7	Environmental Science and Technology	2.29%	4	7.864	23,827	75,022	12.6	Oil spill dispersants: Boon or bane?	2015	105	American Chemical Society
8	FEMS Microbiology Ecology	2.29%	4	3.675	1540	5623	6.5	Hydrocarbon biodegradation by Arctic sea-ice and sub-ice microbial communities during microcosm experiments, Northwest Passage (Nunavut, Canada)	2016	32	Oxford University Press
9	International Biodeterioration and Biodegradation	2.29%	4	4.074	2208	7993	7.9	Exploring the potential of halophilic bacteria from oil terminal environments for biosurfactant production and hydrocarbon degradation under high-salinity conditions	2018	28	Elsevier Ltd.
10	Journal of Hazardous Materials	2.29%	4	9.038	15,501	49,867	13.1	Intrinsic rates of petroleum hydrocarbon biodegradation in Gulf of Mexico intertidal sandy sediments and its enhancement by organic substrates	2013	32	Elsevier B.V.

**Table 3 ijerph-18-01671-t003:** Top ten most productive journals on degradation of hydrocarbons by Antarctic microbial community research with their most cited articles.

Rank	Journal	TP (%)	TP	IF (2019) JCR	TC (2019) JCR	TC (2019) Scopus	Cite Score 2019	The Most Cited Article Based on Source	Year	Cited by	Publisher
1	Frontiers in Microbiology	8.33%	12	4.236	24,092	65,387	6.4	Dynamics of bacterial communities in two unpolluted soils after spiking with phenanthrene: Soil type specific and common responders	2012	60	Frontiers Research Foundation
2	Marine Pollution Bulletin	7.64%	11	4.049	7000	21,687	6.7	Rhamnolipids enhance marine oil spill bioremediation in laboratory system	2013	40	Elsevier Ltd.
3	International Biodeterioration and Biodegradation	4.86%	7	4.074	2208	7993	7.9	Characterization of an alkane-degrading methanogenic enrichment culture from production water of an oil reservoir after 274 days of incubation	2011	80	Elsevier Ltd.
4	Environmental Science and Pollution Research	3.47%	5	3.056	17,158	53,503	4.9	The effect of oil spills on the bacterial diversity and catabolic function in coastal sediments: a case study on the Prestige oil spill	2015	41	Springer Verlag
5	Applied and Environmental Microbiology	2.78%	4	4.016	4761	17,047	7.1	Bacterial communities from shoreline environments (Costa da Morte, northwestern Spain) affected by the Prestige oil spill	2019	104	American Society for Microbiology
6	Journal of Hazardous Materials	2.78%	4	9.038	15,501	49,867	13.1	Recent development in the treatment of oily sludge from petroleum industry: A review	2013	473	Elsevier B.V.
7	Microbial Ecology	2.78%	4	3.356	1208	4531	6.4	Alkane biodegradation genes from chronically polluted subantarctic coastal sediments and their shifts in response to oil exposure	2012	37	Springer Science
8	Environmental Microbiology	2.08%	3	4.933	3320	12,859	9.1	New alk genes detected in Antarctic marine sediments	2009	49	Society for Applied Microbiology
9	Scientific Reports	2.08%	3	3.998	167,821	596,638	7.2	Bacterial population and biodegradation potential in chronically crude oil-contaminated marine sediments are strongly linked to temperature	2015	54	Nature Publishing Group
10	Annals of Microbiology	1.39%	2	1.528	249	1262	2.9	Mangrove sediment, a new source of potential biosurfactant-producing bacteria	2012	18	Springer-Verlag and the University of Milan.

**Table 4 ijerph-18-01671-t004:** Most productive 10 authors that published journal articles between 2009 to 2019 related to degradation of hydrocarbons in the Arctic marine environment by microbial community.

Rank	Author (1st)/Scopus ID	Current Affiliations	h-Index	TP	TC	CPP	Highest Cited Document	Journal	Cited by
1	Brakstad, O.G. (6602165118)	SINTEF Ocean, Trondheim, Norway	22	78	2228	28.56	Estimation of hydrocarbon biodegradation rates in marine environments: A critical review of the Q10 approach (2013 ^b^)	Marine Environmental Research	31
2	Netzer, R. (7004615603)	SINTEF Ocean, Trondheim, Norway	15	30	510	17.00	Microbial community and metagenome dynamics during biodegradation of dispersed oil reveals potential key-players in cold Norwegian seawater (2018 ^c^)	Marine Pollution Bulletin	16
3	Anderson, J.A. (35465962200)	University of Aberdeen, Aberdeen, United Kingdom	50	235	7887	33.56	The variable influence of dispersant on degradation of oil hydrocarbons in subarctic deep-sea sediments at low temperatures (0–5 °C) (2017 ^b^)	Scientific Reports	19
4	Christensen, J.H. (7402503070)	Københavns Universitet, Copenhagen, Denmark	28	117	2507	21.43	Marine biodegradation of crude oil in temperate and Arctic water samples (2015 ^c^)	Journal of Hazardous Materials	20
5	Greer, C. W. (7103169832)	National Research Council Canada, Ottawa ON, Canada	53	203	8652	42.62	Predictable bacterial composition and hydrocarbon degradation in Arctic soils following diesel and nutrient disturbance (2013 ^b^)	ISME Journal	122
6	Hazen, T.C. (7006945153)	The University of Tennessee, Knoxville, Knoxville, United States	55	256	11,597	45.30	Microbial community analysis of a coastal salt marsh affected by the Deepwater Horizon oil spill (2012 ^b^)	PLoS ONE	132
7	King, T.L. (7403270918)	Bedford Institute of Oceanography, Fisheries and Oceans Canada, Dartmouth, Canada	25	93	1621	17.43	Hydrocarbon biodegradation by Arctic sea-ice and sub-ice microbial communities during microcosm experiments, Northwest Passage (Nunavut, Canada) (2016 ^b^)	FEMS Microbiology Ecology	32
8	Ribicic, D. (55597487500)	SINTEF Ocean, Trondheim, Norway	8	15	160	10.67	Microbial community and metagenome dynamics during biodegradation of dispersed oil reveals potential key-players in cold Norwegian seawater (2018 ^a^)	Marine Pollution Bulletin	16
9	Witte, U. F. M. (7003914924)	University of Aberdeen, Aberdeen, United Kingdom	30	85	4438	52.212	The variable influence of dispersant on degradation of oil hydrocarbons in subarctic deep-sea sediments at low temperatures (0–5 °C) (2017 ^c^)	Scientific Reports	19
10	Andersen, G. L. (7202552651)	Lawrence Berkeley National Laboratory, Berkeley, United States	61	158	30,246	191.43	Microbial community analysis of a coastal salt marsh affected by the Deepwater Horizon oil spill (2012 ^b^)	PLoS ONE	133

Abbreviations and notes: TP: total publications; TC: total citations; CPP: citations per publications and representation of superscript, ^a^ first author, ^b^ co-author and ^c^ last author.

**Table 5 ijerph-18-01671-t005:** Most productive 10 authors that published journal articles between 2009 to 2019 related to degradation of hydrocarbons in the Antarctic marine environment by microbial community.

Rank	Author (1st)/Scopus ID	Current Affiliations	h-Index	TP	TC	CPP	Highest Cited Document	Journal	Cited by
1	Cappello, S. (35602948000)	Consiglio Nazionale delle Ricerche, Rome, Italy	25	72	1888	26.22	Bioremediation (bioaugmentation/biostimulation) trials of oil polluted seawater: A mesocosm simulation study (2014 ^c^)	Marine Environmental Research	62
2	Jeon, C.O. (24401297600)	Chung-Ang University, Seoul, South Korea	49	372	8635	23.21	Comparative genomics reveals adaptation by *Alteromonas* sp. SN_2_ to marine tidal-flat conditions: Cold tolerance and aromatic hydrocarbon metabolism (2012 ^c^)	PLoS ONE	55
3	Madsen, E.L. (71018820930	Cornell University, Ithaca, United States	42	111	6180	55.68	The genome of *Polaromonas naphthalenivorans* strain CJ2, isolated from coal tar-contaminated sediment, reveals physiological and metabolic versatility and evolution through extensive horizontal gene transfer (2009 ^c^)	Environmental Microbiology	58
4	Almeida, A. P. (7202913857)	Universidade de Aveiro, Aveiro, Portugal	40	229	5433	23.72	Hydrocarbon contamination and plant species determine the phylogenetic and functional diversity of endophytic degrading bacteria (2014 ^b^)	Molecular Ecology	37
5	Cleary, D.F.R. (7005552581)	Universidade de Aveiro, Aveiro, Portugal	34	129	2918	22.62	Unraveling the interactive effects of climate change and oil contamination on laboratory-simulated estuarine benthic communities (2015 ^b^)	Global Change Biology	20
6	Cunha, A. (57210164924)	Universidade de Aveiro, Aveiro, Portugal	37	147	4118	28.01	Hydrocarbon contamination and plant species determine the phylogenetic and functional diversity of endophytic degrading bacteria (2014 ^c^)	Molecular Ecology	37
7	Dellagnezze, B.M. (36542157500)	Universidade Estadual de Campinas, Campinas, Brazil	7	12	167	13.92	Bioremediation potential of microorganisms derived from petroleum reservoirs (2014 ^a^)	Marine Pollution Bulletin	39
8	Denaro, R. (56072581500)	Istituto di Ricerca sulle Acque, Italy, Monterotondo, Italy	18	39	1307	33.51	Bacterial population and biodegradation potential in chronically crude oil-contaminated marine sediments are strongly linked to temperature (2015 ^b^)	Scientific Reports	54
9	Dionisi, H.M. (6603560555)	Centro Para el Estudio de Sistemas Marinos (CESIMAR), CONICET-CENPAT, Puerto Madryn,	17	42	1984	47.24	Alkane biodegradation genes from chronically polluted subantarctic coastal sediments and their shifts in response to oil exposure (2012 ^b^)	Microbial Ecology	37
10	Gieg, L.M. (6601913918)	University of Calgary, Calgary, Canada	29	63	2583	41.00	Subsurface cycling of nitrogen and anaerobic aromatic hydrocarbon biodegradation revealed by nucleic acid and metabolic biomarkers (2010 ^b^)	Applied and Environmental Microbiology	29

Abbreviations and notes: TP: total publications; TC: total citations; CPP: citations per publications and representation of superscript, ^a^ first author, ^b^ co-author and ^c^ last author.

**Table 6 ijerph-18-01671-t006:** List of oil spills incidents that occurred recently surrounding the Arctic Circle.

Oil Spill Incident	Date	Source of Spill	Incident Site & Country	Type of Oil	Spill Amount (Gallon)	Clean Up Process
Trans Mountain oil spill [38]	14 June 2020	Pipeline	Abbotsford, British Columbia, Canada	Crude oil	50,000	Containment and remediation
Norilsk diesel fuel spill [39]	29 May 2020	Fuel storage tank	Krasnoyarsk Krai, Norilsk, Russia	Diesel fuel	5,440,000	Booms and pump
Tanker truck rollover [40]	21 March 2020	Tanker	Cuyaman River, Santa Maria, United States	Crude oil	4000	Dirt berm, absorbent pads, boom
Keystone Pipeline 2019 spill [41]	29 October 2019	Pipeline	North Dakota, Walsh County, United States	Crude oil	383,000	Backhoes and vacuum trucks
SeaRose FPSO production ship spill [42]	16 November 2018	Vessel flowline	Newfoundland and Labrador, St. John’s Canada	Crude oil	66,000	Monitoring on seabirds
ConocoPhillips, Canada pipeline spill [43]	9 June 2016	Pipeline	Grande Cache, Alberta, Canada	Light petroleum	100,000	Containment

**Table 7 ijerph-18-01671-t007:** List of hydrocarbon-degrading bacteria found in the Arctic.

Type of Hydrocarbon	Microbial Species/Classes	Removal Efficiency	Location	References
Arabian light crude oil and PAHs	Sphingopyxis Flavimaris Pseudoalteromonas Marinobacter antarcticus	17.25–81.98%	Kongsfjorden, Svalbard Islands, Arctic region	[74]
Aromatic (Crude oil)	Magnetospirillum magnetotacticum	95–99% (shorter hydrocarbon chains)	Glacial open fjord Kongsfjorden, at the Research Village in Ny-Alesund; Svalbard Archipelago, Arctic Norway	[81,82,83]
	Sediminicola luteus			
	Microbulbifer pacificus			
	Sphingopyxis flavimaris			
	Thiobacillus thioparus			
Aromatic (Diesel oil)	Cycloclasticus pugetii	75% (longer hydrocarbon chains)		
	Novosphingobium nitrogenifigens			
	Pibocella ponti			
	Magnetospirillum gryphiswaldense			
PAHs	Cycloclasticusa Pseudomonasa	No information	Contaminated sediments, Makarov Basin, Arctic	[92]
Crude oil	Alcanivorax borkumensis	No information	Barents Sea (Russia)	[93]

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
