# Peer review of "A Review and Bibliometric Analysis on Applications of Microbial Degradation of Hydrocarbon Contaminants in Arctic Marine Environment at Metagenomic and Enzymatic Levels"

_ijerph, 2021, doi:10.3390/ijerph18041671_

Round 1
Reviewer 1 Report
This is a review paper on biodegradation of hydrocarbons in cold environment with references to word in the Arctic on the last 10 years. Authors indicated in the Introduction that they focused on diesel spill but this focus was not clear in the rest of the paper. Why focusing on diesel only? This is not a bigger job to include all types of oil spills. Otherwise, the work is well done and would be useful to students starting their research in that field and also to professors for academic purposes.
Author Response
Comment
This is a review paper on biodegradation of hydrocarbons in cold environment with references to word in the Arctic on the last 10 years. Authors indicated in the Introduction that they focused on diesel spill but this focus was not clear in the rest of the paper. Why focusing on diesel only? This is not a bigger job to include all types of oil spills. Otherwise, the work is well done and would be useful to students starting their research in that field and also to professors for academic purposes.
Answer: Different types of fuel oil especially used in the Arctic marine environment was discussed in the Introduction section. (Page 2, Paragraph 1, Line 45-58)

Reviewer 2 Report
Thank you for the opportunity to review the article entitled A Review and Bibliometric Analysis on Applications of Microbial Degradation of Hydrocarbon Contaminants in Arctic Marine Environment at Metagenomic and Enzymatic Levels. In my opinion the article is a very valuable literature item in the scope of the conducted review and it will be also a valuable literature item for scientists conducting research in this field. However, I have considerable doubts whether such bibliometric analysis falls within the scope of the International Journal of Environmental Research and Public Health. I will leave this doubt to the journal's Editor-in-Chief for evaluation. I noticed a few shortcomings that should be corrected before the publication of this article:
- I propose to delete (a), (b) and (c) in the title of section 3. Numbering can be used for subsections.
- Figure 1 should be provided in better quality. The order of citing the references in the caption should be [21,23], not [23,21].
- Have the authors also tried to find research papers that are not indexed in Scopus? Did the authors recognize the publication outside the Scopus base?
- Figures 3a and 3b are illegible.
Author Response
Comment 1
I propose to delete (a), (b) and (c) in the title of section 3. Numbering can be used for subsections.
Answer: The sub-topics in Section 3 changed to numbering and rearranged after bibliometric analysis (Page 2, Section 2).
Previously as:
3 (a) changed to 3.1 (Line 33)
3 (b) changed to 3.2 (Line 44)
3 (c) changed to 3.3 (Line 57)
Comment 2
Figure 1 should be provided in better quality. The order of citing the references in the caption should be [21,23], not [23,21].
Answer: Figure 1 renamed as Figure 7 (Line 15) and provided in high resolution.Order of citing references were amended throughout the manuscript. (Page 22).
Comment 3
Have the authors also tried to find research papers that are not indexed in Scopus? Did the authors recognize the publication outside the Scopus base?
Answer: As discussed in Section 2 (Page 3, Line 125-134).
Comment 4
Figures 3a and 3b are illegible.
Answer: The number of keywords occurred were reduced and the figure was reconstructed. Only keywords closely related to the topic was chosen for network analysis using VOS viewer. Page 5-6. Figure 3 (a & b) renamed as Figure 2 (Line 188 & 190).
Reviewer 3 Report
Comments:
- Why do you think it is necessary to review the literature on this subject? Who can it help, beyond a description of past activity?
- Could another methodology have been applied? That is, this method has a series of limitations (it would be convenient to explain them), so that a combination with another, for example, a qualitative methodology could help to clarify the characteristics of this scientific activity.
- Consider updating the bibliography:
https://doi.org/10.1016/j.marpol.2017.10.022
https://doi.org/10.3390/pr8050631
- What does sections 2 and 3 contribute to in a bibliometric analysis paper?
- Figure 1 does not have a good definition in the PDF. Is it possible to improve it?
- I note that they have only taken into account the articles. What studies indicate that selecting these in the sample is positive? Why not include preprints or book chapters, which also have peer review?
- In figure 5 there are subject areas with 0%. Include 2 decimals to make sense.
- Properly discuss journals by publisher so that the column in Table 3 makes sense.
Author Response
Comment 1
Why do you think it is necessary to review the literature on this subject? Who can it help, beyond a description of past activity?
Answer: The Introduction section (paragraph 2 & 3) was revised by highlighting the focus of this study clearly. The purpose to review the literature on this subject and who can it help also described in paragraph 2 and 3 (Line 73-98) also mentioned in Section 3 (Line 3-7, page 22).
The introduction revised as following flow:
- First, a brief overview and the aim of bibliometrics were discussed. The bibliometrics describes the application of microbial communities in hydrocarbons degradation focused in both the Arctic and Antarctic marine environments.
- Based on the research trends analysis, a literature review explains focused on microbial community hydrocarbons degradation in the Arctic marine environment
Comment 2
Could another methodology have been applied? That is, this method has a series of limitations (it would be convenient to explain them), so that a combination with another, for example, a qualitative methodology could help to clarify the characteristics of this scientific activity.
Answer: The series of limitations were discussed in new section, Section 2.7, Page 18.
Comment 3
Consider updating the bibliography:
https://doi.org/10.1016/j.marpol.2017.10.022
https://doi.org/10.3390/pr8050631
Answer: As cited in Section 2.7 (Page 18).
https://doi.org/10.1016/j.marpol.2017.10.022 [31]
https://doi.org/10.3390/pr8050631 [29]
Comment 4
What does sections 2 and 3 contribute to in a bibliometric analysis paper?
Answer: Page 2, Page 3-22
- Previous Section 2 was revised, and the types of oils or fuel related to Arctic marine pollution was mentioned in the Introduction (Line 45-58).
- Section 2 replaced with bibliometric analysis (previously in Section 4)
- The manuscript revised to highlight the bibliometric analysis (current Section 2) first then followed by a literature review (current Section 3-5) focused on the microbial hydrocarbon degradation in Arctic marine.
Comment 5
Figure 1 does not have a good definition in the PDF. Is it possible to improve it?
Answer: Figure 1 renamed as Figure 7 and provided in high resolution. (Page 22)
Comment 6
I note that they have only taken into account the articles. What studies indicate that selecting these in the sample is positive? Why not include preprints or book chapters, which also have peer review?
Answer: The reason for selecting articles only from the database were described with citations (Paragraph 3, Line 125-134). Page 3
Comment 7
In figure 5 there are subject areas with 0%. Include 2 decimals to make sense.
Answer: Figure 5 was modified with 2 decimals points and rearranged as Figure 4 (Line 225). Page 8.
Comment 8
Properly discuss journals by publisher so that the column in Table 3 makes sense.
Answer: Discussion added for journals’ publisher as in Table 2 and 3 in Section 2.5 (Paragraph 1, Line 276-279). Page 11-12.
Round 2
Reviewer 3 Report
Comments:
- Figure 2 looks blurry. Please attach it with higher quality.
- In lines 158-159: "The number of keywords corelates was reduced to allow a clear representations of network mapping". Why did they make this decision? It would have been better to include more terms to see how they are related. What does color grouping mean?
- It is possible to determine with which terms one more term is linked (with what theme),
- With what end is the link greater for each of the terms? This would help us establish sub-themes.
- the results of section 2.2 need a discussion?
- It is convenient to establish a relationship between the affiliations and the authors and countries. What links are found? Have they been the same links throughout the analyzed period or have there been differences?
- Table 4. It is also convenient to determine whether or not the article in question was relevant to the subject of study. The citations do not indicate this relevance.
Author Response
Comment 1
Figure 2 looks blurry. Please attach it with higher quality.
Answer: Figure 2 provided in higher resolution. (Page 6)
Comment 2
In lines 158-159: "The number of keywords correlates was reduced to allow a clear representations of network mapping". Why did they make this decision? It would have been better to include more terms to see how they are related. What does colour grouping mean?
Answer: More terms were included (Figure 2) and the discussion has been revised. (Line 157-161) (Page 4).
Comment 3
It is possible to determine with which terms one more term is linked (with what theme)
Answer: As discussed in Line 166-182, 193-213 (Page 4-5)
Comment 4
With what end is the link greater for each of the terms? This would help us establish sub-themes.
Answer: The discussion has been revised. (Line 157-161, 199-203) (Page 4)
Comment 5
the results of section 2.2 need a discussion?
Answer: Discussion has been added (Line 228-230 and 231-232) Page 7.
Comment 6
It is convenient to establish a relationship between the affiliations and the authors and countries. What links are found? Have they been the same links throughout the analyzed period or have there been differences?
Answer: A brief discussion provided in the Line 379-387 (Page 16)
Comment 7
Table 4. It is also convenient to determine whether the article in question was relevant to the subject of study. The citations do not indicate this relevance.
Answer: Table 4 and 5 were removed. Considering the length of manuscript and Table 4 and 5 are not significant to the topic of the study. (Page 17-18) Numbering of table has been updated.
